# Fertility Preservation in Cervical Cancer—Treatment Strategies and Indications

**Lina Salman [1] and Allan Covens [1,2,*]**

1  Division of Gynecologic Oncology, Department of Obstetrics & Gynecology, University of Toronto, Toronto, ON M5G 2M9, Canada; lina.salman@uhn.ca
2  Division of Gynecologic Oncology, Odette Cancer Centre, Sunnybrook Health Sciences Centre, Toronto, ON M4N 3M5, Canada
*  Correspondence: al.covens@sunnybrook.ca

**Abstract:** Cervical cancer is frequently diagnosed in women during their reproductive years, and fertility preservation is an essential part of their cancer treatment. In highly selected patients with early stage, low-risk cervical cancer and a tumor size ≤ 2 cm, several treatment strategies can be offered for patients wishing to preserve fertility, including radical/simple trachelectomy or conization with pelvic lymph node assessment. Trachelectomy can be performed through a vaginal, abdominal, or minimally invasive approach and has been shown to have an equivalent oncologic outcome compared to radical hysterectomy. All surgical approaches for radical trachelectomy seem to have excellent survival with comparable oncologic outcomes. Nevertheless, patients undergoing vaginal trachelectomy have better obstetric outcomes compared to the other routes. In patients with larger tumors (2–4 cm), neoadjuvant chemotherapy followed by fertility-sparing surgery is an alternative option. Several chemotherapy regimens have been used for this indication, with a pathologic complete response rate of 17–73%. For locally advanced diseases that require radical hysterectomy or primary chemoradiation, fertility preservation can be performed using oocyte, embryo, or ovarian tissue cryopreservation, as well as ovarian transposition. For these patients, future pregnancy is possible through surrogacy. In addition to fertility preservation, ovarian transposition, where the ovaries are repositioned outside of the radiation field, is performed to maintain ovarian hormonal function and prevent premature ovarian failure. In summary, fertility-preservation treatment strategies for patients with early stage cervical cancer are continuously evolving, and less radical surgeries are becoming more acceptable. Additional and ongoing evidence is helping determine the impact of conservative procedures on oncologic and obstetric outcomes in these patients.

**Keywords:** cervical cancer; fertility preservation; trachelectomy; ovarian transposition

## 1. Introduction

Cervical cancer is the fourth most common cancer in women worldwide [1].

It is frequently diagnosed in women aged 35–44, and it is the second leading cause of cancer death in women aged 20–39 [2]. Treatment of cervical cancer is based on the stage of the disease [3]. The standard treatment for patients with early stage disease (stage IA2-IB1) is/has been radical hysterectomy with pelvic lymph node dissection, while patients with locally advanced and metastatic disease are treated with primary radiation therapy (RT) +/− systemic treatment [3,4]. As 37% of patients with newly diagnosed cervical cancer are under the age of 45 [5], fertility preservation treatment options are often desired. Surgical treatment modalities include radical and simple trachelectomy as well as cervical conization. In certain cases of locally advanced disease where uterine preservation is not an option, fertility preservation can be maintained through assisted-reproduction technologies (ART) and ovarian transposition, which also have the advantage of preserving ovarian hormonal function [6–8]. In this review, we present fertility preservation treatment strategies for early

stage cervical cancer, including indications, oncologic, and obstetric outcomes. In addition, we present evidence supporting conservative management of these patients.

## 2. Radical Trachelectomy with Lymph Node Assessment

Selected patients with early stage cervical cancer wishing to preserve fertility might be eligible for trachelectomy [9]. Radical trachelectomy was first described by Dargent in 1987 [10], and it consists of the removal of the cervix, vaginal margins, and parametria. Historical indications for trachelectomy included stage IA1 with lymphovascular space invasion (LVSI), stage IA2-IB1 (tumor size ≤ 2 cm), negative nodal metastasis, and the absence of deep stromal invasion [3,11]. Excluding nodal disease prior to fertility preservation treatment is crucial, as positive lymph nodes are a poor prognostic factor and can determine the appropriate treatment [12]. The feasibility of sentinel lymph node mapping led to a paradigm shift in lymph node assessment in cervical cancer [3]. According to the NCCN guidelines, sentinel lymph node biopsy can replace pelvic lymph node dissection in FIGO 2018 stage IA1 with LVSI and stage IA2-IB1 [3,13], whereas in the 2023 ESGO guidelines, pelvic lymph node dissection should be performed in stage IB1 with negative sentinel nodes frozen [4]. Although data from the prospective SENTI-COL I and SENTICOL II studies showed that omitting full pelvic lymphadenectomy for patients with stage IA-IIA with bilateral negative sentinel lymph nodes does not seem to be associated with an increased recurrence rate, evidence on the accuracy and oncologic safety of sentinel lymph node biopsy in cervical cancer is still evolving, and additional prospective data are needed [14]. Nica et al. evaluated the outcome of patients with early stage cervical cancer (tumor size ≤ 2 cm) undergoing cervical conization with laparoscopic sentinel lymph node assessment [15]. Of the 44 patients included in the analysis, 93% had negative sentinel nodes and did not require further nodal procedures, while 6.8% of patients had micrometastases detected in the sentinel nodes and underwent ipsilateral lymphadenectomy. All the remaining non-sentinel lymph nodes were negative. After a median follow-up of 44 months, no recurrences were documented. Another retrospective study included 36 patients with stage IA2-IBI cervical cancer [16]. All patients underwent FSS and laparoscopic sentinel lymph node dissections. Of those, 50% underwent sentinel lymph node dissection alone, and 50% underwent sentinel lymph node followed by full pelvic lymph node dissection. In total, there were four recurrences: two in the sentinel and two in the pelvic lymphadenectomy groups. The results of these studies are promising, but larger prospective studies are required to assess the accuracy of sentinel lymph node dissection in early stage cervical cancer and to evaluate the safety of minimally invasive surgeries for lymph node assessment.

Radical trachelectomy originally combined vaginal approaches. Since then, the procedure has been modified and is currently performed via a vaginal, abdominal, or minimally invasive approach, based on the surgeon's preference and experience [17]. Several studies evaluated the oncologic outcome of radical trachelectomy and found it to have comparable survival and recurrence rates compared to radical hysterectomy [18–20], making it a safe alternative for patients wishing to preserve fertility.

The question of whether one surgical approach is superior to the other has been evaluated in several retrospective studies. A systematic review by Smith et al. compared the surgical, oncologic, and obstetric outcomes of radical trachelectomy according to the surgical route: vaginal, abdominal, and laparoscopic [17]. Out of 2566 patients included in the analysis, 75% had stage IB1 (tumor size < 2 cm). The majority of patients underwent vaginal radical trachelectomy (58%), 37% had abdominal procedures, and only 4.7% underwent laparoscopic trachelectomy. The vaginal approach had a shorter median operative time compared to the abdominal and laparoscopic routes, and had lower rates of positive margins. The post-operative pregnancy rate was found to be highest in the vaginal approach (38%) compared to abdominal and laparoscopy (10% and 9%, respectively), in addition to lower rates of preterm delivery. It should be noted that the group undergoing vaginal trachelectomy had a longer follow-up time, which might explain the higher pregnancies

reported. While this review included a large number of patients, it is based on published retrospective data; therefore, selection and publication biases should be taken into account when interpreting these findings.

Since the publication of the Laparoscopic Approach to Cervical Cancer (LACC) trial [21], the safety of minimally invasive surgery (MIS) for radical trachelectomy has raised some concerns. The LACC trial was a randomized controlled trial that evaluated disease-free survival in patients undergoing MIS versus open radical hysterectomy for early stage cervical cancer. The results demonstrated increased recurrence and death rates in patients undergoing radical hysterectomy via the MIS approach in patients with stage IB2, as these comprised the majority of the study cohort. Prospective data on MIS versus open surgeries for stage IA2/IB1 cervical cancer are limited, as is the safety of MIS for lymph node assessment in these patients. A retrospective study using the National Cancer Database evaluated the trends, characteristics, and survival outcomes of patients with stage IA2-IB cervical cancer undergoing radical trachelectomy via MIS versus laparotomy [22]. Of the 246 people included in the study, 144 underwent surgery via the MIS approach. Patients undergoing vaginal trachelectomy were excluded. The authors found a significant increase in using the MIS approach throughout the years (increasing from 29% in 2010 to 75% in 2015). Death events were 7.6% in the laparotomy group compared to 3.5% in the MIS ($p = 0.025$). The absolute number of events in the study was very low, and the authors conclude that although no survival difference was found, the effect of MIS radical trachelectomy on survival remains unknown.

A further comparison of MIS versus open radical trachelectomy was published by the International Radical Trachelectomy Assessment (IRTA) [23]. In this retrospective study, 646 patients were included in the analysis. 358 underwent open surgery, and 288 underwent MIS. At 4.5 years, 4.8% had a recurrence in the open surgery and 6.3% in the MIS, but this was not statistically significant. In addition, there was no difference in overall survival between groups (99.2% in open surgery vs. 99.0% in MIS). As this is not a common procedure, the feasibility of performing a randomized clinical trial to evaluate survival outcomes in MIS versus open trachelectomy is unlikely, and clinical practice will be based on the best available evidence.

Regardless of the surgical approach, the radicality of the procedure is continuously being refined for more conservative surgeries. Based on tumor factors, patients may undergo simple trachelectomy or cervical conization rather than radical trachelectomy with similar oncologic outcomes [24].

## 3. Simple Trachelectomy and Cone Biopsy

The rationale for performing more conservative surgeries is supported by the results of several studies evaluating the risk of parametrial involvement in patients with early stage cervical cancer who have favorable pathologic features (tumor $\leq$ 2 cm, depth of invasion $\leq$ 10 mm, and negative pelvic nodes) [25,26]. In this well-defined cohort of patients, the risk of parametrial involvement is lower than <1%, putting the benefit of removing the parametria in doubt. While cervical conization is typically performed to treat high-grade premalignant lesions of the cervix [27], the above evidence led to studies evaluating the performance of cone biopsy as a treatment for cervical cancer. The ConCerv trial was the first prospective study to evaluate the feasibility and oncologic outcomes of conization alone or simple hysterectomy in early stage low-risk cervical cancer [28]. In this study, they included patients with FIGO 2009 stage IA2-IB1 cervical cancer [29] who meet the following criteria: squamous cell or adenocarcinoma, tumor size $\leq$ 2 cm, no LVSI, depth of invasion $\leq$ 10 mm, negative imaging for metastatic disease, and negative conization margins. Patients were allowed to undergo a repeated cone if the first one had positive margins. Patients desiring fertility underwent a second conization with pelvic lymph node assessment, and those not desiring fertility preservation underwent a simple hysterectomy with pelvic lymph node assessment. In total, 44 patients were enrolled in the fertility preservation arm. Of those, two patients had positive lymph nodes, and one patient had

recurrent disease. The one patient with recurrent disease had a stromal invasion of 13 mm on the first conization with positive margins. Repeated conization was negative for cancer, but margins were positive for high-grade dysplasia. This had led the investigators to amend the protocol and exclude patients with positive margins not only for invasive cancer but also for intra-epithelial neoplasia. Data from Bogani et al. on 32 patients with FIGO 2018 stage IA2, IB1, and IB2 undergoing conization with pelvic lymph node assessment showed 5-year disease-free survival and overall survival to be 94% and 97%, respectively. Another published study by Plante et al. evaluated the obstetric and oncologic outcomes of simple vaginal trachelectomy/conization in patients with low-risk, early stage cervical cancer. The 5-year progression-free survival and overall survival were 97.9% and 97.6%, respectively [30].

A systematic review looking into obstetrics and oncologic outcomes following fertility preservation treatment for early cervical cancer included 347 cases that underwent conization [31]. In this group, the recurrence rate was 0.4%, and the pregnancy rate was 36.1% with no death events. A more recent systematic review published by Nezhat et al. evaluated reproductive and oncologic outcomes after fertility-sparing surgery (FSS) for stage IA1-IB1 cervical outcomes [32]. They included patients who underwent conization/simple trachelectomy, or radical trachelectomy via different surgical approaches. Of the 3044 patients included, the pregnancy rate was 55.4% in patients attempting to conceive, with the highest clinical pregnancy rate after vaginal trachelectomy (67.5%). After a median follow-up of 39.7 months, the mean cancer recurrence rate was 3.2%, and the cancer death rate was 0.6%. These data highlight the excellent oncologic outcome and safety of performing these procedures in these patients.

Another aspect that should be considered when choosing a treatment modality is the impact of such treatment on quality of life. The Gynecologic Oncology Group (GOG)-0278 is a phase I/II study evaluating physical function and quality of life in patients with cervical cancer stage IA1 with LVSI and IA2-IB1 (≤2 cm) who underwent simple hysterectomy or cone biopsy with pelvic lymphadenectomy (ClinicalTrials.gov/NCT01649089). This study will look into urinary, gastrointestinal, and sexual function following non-radical surgery. The study completed accrual, and results are anticipated to be presented in early 2024.

## 4. Oncologic Safety of Fertility-Preservation Surgeries in Different Histologic Types

Squamous cell carcinoma (SCC) is the most common histological type of cervical cancer, followed by adenocarcinoma, which accounts for approximately 25% of all cases [33]. The incidence of adenocarcinoma in developed countries has increased in the last decade, whereas the incidence of SCC has been decreasing [34]. Studies on patients with early stage cervical cancer undergoing definitive treatment showed conflicting results regarding the oncologic outcome of adenocarcinoma compared to SCC [35–37]. Concerns were raised regarding the oncologic safety of FSS in patients with adenocarcinoma or adenosquamous. Zusterzeel et al. looked at recurrence risk following radical vaginal trachelectomy in early stage cervical cancer [38]. In this retrospective study of 132 patients, 72% had SCC, 24.2% had adenocarcinoma, and 3.8% had adenosquamous. The majority of patients in this study had FIGO 2009 stage IB1 disease. The overall recurrence rate was 6.8%, with a median time to recurrence of 21 months. Adenosquamous carcinoma had the highest recurrence rate of 20%, followed by 12.5% for adenocarcinoma and 4.2% for SCC. Additional studies failed to show a difference in oncologic outcome in adenocarcinoma compared to SCC for patients undergoing radical trachelectomy [39,40]. A recently published multicenter study evaluated risk factors for recurrence in patients with early stage cervical cancer treated with FSS [41]. They included 733 patients who underwent any type of FSS. The majority of patients (70%) had SCC, and 24% had adenocarcinoma. A total of 49% of patients had FIGO 2018 stage IB1 disease. After a median follow-up of 72 months, histologic subtype was not found to be a risk factor for recurrence. In fact, the only significant risk factor was tumor size >2 cm.

While some histological types have a worse prognosis than SCC, it is not evident that FSS increases the risk of recurrence. Although additional adjuvant therapy might be indicated based on pathology to decrease the risk of recurrence, FSS should not be a contraindication to its administration.

## 5. Neoadjuvant Chemotherapy

Neoadjuvant chemotherapy (NACT) is an alternative option for patients with bulky cervical cancer (tumor size > 2 cm) wishing to preserve fertility. The rationale for administering chemotherapy is to shrink the tumor and make an FSS feasible. Different chemotherapy regimens were studied in small case series, including paclitaxel/cisplatin/ifosfamide (TIP) [42,43], paclitaxel/cisplatin [44], and carboplatin/paclitaxel [45], with varying numbers of cycles (Table 1).

**Table 1.** Summary of several studies evaluating neoadjuvant chemotherapy followed by fertility-sparing surgery for cervical cancer.

| Study | # of Patients | Stage of Disease [a] | Chemotherapy Regimen | # of Chemo Cycles | Complete Pathologic Response to NACT | FSS after NACT |
|---|---|---|---|---|---|---|
| Maneo et al., 2008 [43] | 21 | IB1 (tumor size < 3 cm) | Cisplatin 75 mg/m$^2$, paclitaxel 175 mg/m$^2$, and ifosfamide 5 g/m$^2$ | 3 | 24% | PLND + conization |
| Vercellino et al., 2012 [46] | 18 | IB1-IB2 (tumor size 2–5 cm) | Ifosfamide 5 g/m$^2$, cisplatin 100 mg/m$^2$, and paclitaxel 200 mg/m$^2$ | 2–3 | 17% | PLND + RVT |
| Lanowska et al., 2014 [47] | 18 | IB1-IB2 (tumor size 2–5 cm) | Ifosfamide 5 g/m$^2$, cisplatin 100 mg/m$^2$, and paclitaxel 200 mg/m$^2$ | 2–3 | 50% | PLND + RVT |
| Robova et al., 2014 [48] | 28 | IB1-IB2 (tumor size 1.5–4 cm) | 1.Cisplatin 75 mg/m$^2$ and ifosfamide (2 g/m$^2$) **OR** 2. Cisplatin (75 mg/m$^2$) and doxorubicine (35 mg/m$^2$) | 3 | 21% | PLND + SVT |
| Salihi et al., 2015 [49] | 11 | IB1-IB2 (tumor size 1.2–5.2 cm) | 1.Paclitaxel 90 mg/m$^2$ and carboplatin AUC 4 **OR** 2. Ifosfamide 5 g/m$^2$, cisplatin 75 mg/m$^2$, and paclitaxel 175 mg/m$^2$ **OR** 3. Dose-dense paclitaxel 60 mg/m$^2$ and carboplatin AUC 2 | 3–9 | 73% | Conization |
| Tesfai et al., 2020 [45] | 19 | IB1-IIA (tumor size 3.5–6 cm) | Weekly cisplatin 70 mg/m$^2$ and paclitaxel 70 mg/m$^2$ | 6 | 47% | ART |
| Zusterzeel et al., 2020 [50] | 18 | IB2 [b] (tumor size 2.2–4 cm) | Weekly cisplatin 70 mg/m$^2$ and paclitaxel 70 mg/m$^2$ | 6 | 39% | RVT |

[a] According to 2009 FIGO staging; [b] according to 2018 FIGO staging. #—Number; NACT—Neoadjuvant Chemotherapy; FSS—Fertility-Sparing Surgery; PLND—Pelvic Lymph Node Dissection; RVT—Radical Vaginal Trachelectomy; SVT—Simple Vaginal Trachelectomy; and ART—Abdominal Radical Trachelectomy.

The results of several case series investigating the oncologic and obstetric outcomes of NACT followed by fertility preservation surgery in patients with tumor sizes 2–4 cm have been summarized in a systematic review by Gwacham et al. [51]. All patients included in this review (n = 114) had FIGO 2018 stage IB2 cervical cancer. The most common chemotherapy regimen was TIP (89.5% of patients). Pelvic lymphadenectomy was performed in 49% of patients prior to starting NACT, whereas 51% underwent NACT without nodal assessment. FSS was performed on 99.1% of patients. The most common procedure performed was radical vaginal trachelectomy (40.7%). The response to treatment was high, with a complete pathologic response reported to be 39.5% and a partial response of 45.6%. As for obstetric outcomes, 69.4% had full-term delivery, 9.7% had preterm delivery, and 16.1% had miscarriages. Although these data are obtained from small retrospective studies, these findings are promising. As mentioned earlier, different chemotherapy regimens were used, and it is not clear whether one regimen is superior to the other. In addition, there

was no consistency between studies in the timing of performing lymph node dissection. While some studies performed lymph node dissection after NACT, one would argue that it should be performed prior to starting NACT, as positive nodes are associated with a poor prognosis and these patients would require adjuvant treatment with chemotherapy and radiation [52].

The outcome of NACT, followed by FSS, is currently being investigated by the CoNtESSA trial. This is a prospective multi-center trial evaluating NACT followed by FSS for premenopausal patients with cervical cancer, FIGO 2018 stage IB2, wishing to preserve fertility [53]. Participants are treated with NACT, consisting of platinum-based chemotherapy (cisplatin or carboplatin) with paclitaxel. Those with a complete/partial response will undergo fertility-sparing surgery. The primary end point of this study is to assess the rate of functional uterus defined as successful fertility-sparing surgery and no adjuvant therapy. The study is recruiting, and results are expected in 2025.

## 6. Ovarian Transposition

In patients with locally advanced cervical cancer, treatment includes external beam radiation (EBRT) +/− brachytherapy +/− chemotherapy [3,54]. The standard dose used in EBRT for cervical cancer is lethal to the ovaries and leads to ovarian failure [55]. In premenopausal patients, this can result in a post-menopausal state with its associated symptoms and manifestations such as vasomotor symptoms, urogenital atrophy, osteoporosis, and long-term cardiovascular complications [56,57]. In patients receiving EBRT, ovarian transposition (OT) can be offered prior to treatment initiation to preserve ovarian function in addition to fertility preservation. In this procedure, the ovaries are transposed laterally, well above the pelvic brim, avoiding tension or torsion of the gonadal vessels [58,59]. When OT is considered, it should be performed as soon as possible and with a minimally invasive approach to enhance recovery, as a longer duration of time from diagnosis to treatment negatively affects prognosis [60]. In a systemic review by Buonomo et al. [61], looking into the outcomes of 1377 patients with cervical cancer undergoing OT followed by RT, it was found that ovarian function was preserved in 61.7% (range 16.6–100%). Several factors could explain the low rate of ovarian function preservation noted in that study: First, there are surgical techniques, as the transposed ovaries might not be situated far enough from the radiation borders, therefore exposing the ovaries to significant amounts of scatter radiation. Second, the patient's age at the time of the procedure plays an important factor. Although this procedure is performed in premenopausal patients, the more advanced the patient's age, the higher the likelihood of ovarian failure [62]. Smaller doses of exposure to radiation in these patients can lead to ovarian insufficiency as the effective sterilizing radiation dose decreases with increasing age [63].

In addition to preserving hormonal function, ovarian transposition plays a role in fertility preservation, and there have been reports of successful pregnancies following this procedure [6]. In patients with preserved ovarian function, ovulation induction and transabdominal oocyte retrieval can be performed, with successful pregnancies reported through surrogacy [61,64].

## 7. Cryopreservation of Oocytes, Embryos and Ovarian Tissue

For young patients who are not eligible for any of the fertility preservation options discussed above, it is important to refer them to fertility specialists to discuss other options using ART. Fertility preservation can be performed through mature oocyte cryopreservation, embryo cryopreservation, or, in cases where chemotherapy cannot be delayed, ovarian tissue cryopreservation (OTC) [7] (Table 2). Controlled ovarian hyperstimulation with gonadotropins, followed by oocyte retrieval and cryopreservation of oocytes or embryos, can be performed before initiating gonadotoxic treatments [8]. The stimulation can have a "random start" regardless of the phase of the menstrual cycle, which facilitates the process without impacting the quality or number of retrieved oocytes [65]. A study looking at the long-term reproductive outcome of controlled ovarian stimulation in patients with

gynecologic malignancies found that 17 out of 68 patients (25%) returned to the clinic to claim their oocytes/embryos in a median time of 36 months. Out of this sample, the successful livebirth rate was 58.8% [66]. While oocyte and embryo cryopreservation are well-established techniques for fertility preservation, OTC is considered an innovative technique [67]. In OTC, the cortex of harvested ovarian tissue is separated and cryopreserved. Once gonadotoxic treatment is completed, the ovarian tissue can be thawed and transplanted back to the patient to regain ovarian hormonal function and fertility [68]. The site of transplant can be on the remaining ovary, pelvic side walls, subcutaneously, or intramuscularly [67]. In a systematic review and individual patient data meta-analysis of ovarian tissue transplants, 87 studies and 735 women were included [68]. In this review, most patients underwent ovarian transplant via laparoscopy, either to the remaining ovary or pelvic side wall/peritoneum. Pooled rates for pregnancy were 37%, and the live birth rate was 28%. The median time of graft function was 2.5 years (range 0.7–5 years). In the cohort included in this study, it was not surprising but worth noting that no pregnancies were achieved in patients with cervical cancer. This has been demonstrated in other studies showing that patients with cervical cancer have a lower chance of pregnancy compared to other types of cancer [69]. This is secondary to radiation-induced uterine fibrosis [70].

**Table 2.** Comparison of different artificial reproductive technologies used in fertility preservation for cervical cancer.

| Artificial Reproductive Technology | Indication in Cervical Cancer | Advantages | Reported Pregnancy Rate | Reported Livebirth Rate |
|---|---|---|---|---|
| Ovarian transposition | Prior to pelvic radiation | Preserves ovarian hormonal function | 75–89% [59] | - |
| Oocyte and embryo cryopreservation | Prior to systemic chemotherapy | Well-established technique with high success rate | - | 58.8% [66] |
| Ovarian tissue cryopreservation | Prior to systemic chemotherapy | Can be performed in any patient irrespective of age, without delays in cancer treatment | 37% [68] | 28% [68] |

While this seems to be a promising option for fertility preservation, there are some concerns regarding this approach, mainly the risk of recurrence. Ovarian tissue from cancer patients may have microscopic disease, and auto-transplantation of this tissue can theoretically lead to cancer recurrence [71]. Although the harvested tissue is examined for cancer cells, the strip of tissue examined is not used for cryopreservation, and one could argue that ovarian tissue that is actually cryopreserved has cancer cells. More research is required in this field, as there is no consensus regarding the size of ovarian tissue used for auto-transplants or the ideal site of transplantation [68].

## 8. Summary

Several fertility preservation modalities are available for patients with early stage cervical cancer. Radical trachelectomy has an excellent survival outcome with low recurrence rates when compared to radical hysterectomy, regardless of the surgical approach. The low risk of parametrial involvement in early stage disease justifies the performance of less radical procedures such as simple trachelectomy and cone biopsy for highly selected patients.

For patients with large tumors not eligible for primary surgery, NACT is associated with a good response, making an FSS more feasible.

In cases where uterine preservation is not possible, fertility preservation can be performed using cryopreservation of oocytes, embryos, and ovarian tissues, as well as ovarian transposition. These methods enable patients to have future fertility through ART and surrogacy.

Great advances have been made in treating cervical cancer; however, evidence on treatment options for fertility preservation should be interpreted with caution as it is extrapolated from retrospective and prospective single-arm studies. Since randomized clinical trials are unlikely to be feasible due to the rarity of these procedures, multi-institutional collaboration is needed to continuously evaluate patients' outcomes, especially in an era of less radicality.

**Author Contributions:** L.S.—conceptualization; methodology; resources; writing—original draft preparation; writing—review and editing; and visualization. A.C.—conceptualization; methodology; resources; writing—review and editing; visualization; and supervision. All authors have read and agreed to the published version of the manuscript.

**Funding:** This research received no external funding.

**Conflicts of Interest:** The authors declare no conflicts of interest.

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
