# Peer review of "Fertility Preservation in Cervical Cancer—Treatment Strategies and Indications"

_curroncol, doi:10.3390/curroncol31010019_

Round 1

Reviewer 1 Report

Comments and Suggestions for Authors

This is a well written comprehensive review on various fertility sparing techniques in cervix cancer. 

Minor flaws:

1. In the introduction the authors state that according to the ESGO guidelines (2019) stage IA2/IB1 is preferably treated by radical hysterectomy. However, this ESGO guideline mentions that stage IA2 can be treated without parametrectomy. Therefore I suggest that the authors correct this. Please also note that an updated version of this guideline was published this year (ESGO 2023)

2. I miss the discussion on the oncological safety of (radical) trachelectomy in relation to histotype. There are indications that non squamous histology might have a negative impact on recurrence and survival after FSS (Mangler et al 2014; Zusterzeel et al 2016) . To be complete I would recommend to also discuss this issue.

Author Response

This is a well written comprehensive review on various fertility sparing techniques in cervix cancer. 

Minor flaws:

  1. In the introduction the authors state that according to the ESGO guidelines (2019) stage IA2/IB1 is preferably treated by radical hysterectomy. However, this ESGO guideline mentions that stage IA2 can be treated without parametrectomy. Therefore I suggest that the authors correct this. Please also note that an updated version of this guideline was published this year (ESGO 2023)

Authors’ Response: We thank the reviewer for this comment. As the guidelines are continuously changing and treatment for cervical cancer is becoming less radical, we worded the sentence as “the standard treatment is/has been…” to emphasize the evolution of changes in these guidelines. The NCCN guideline recommend modified radical hysterectomy for stage IA2, therefore we kept the sentence as it is but changed the references to the NCCN and the 2023 ESGO guidelines. (See lines 37-39 in the revised paper)

  1. I miss the discussion on the oncological safety of (radical) trachelectomy in relation to histotype. There are indications that non squamous histology might have a negative impact on recurrence and survival after FSS (Mangler et al 2014; Zusterzeel et al 2016) . To be complete I would recommend to also discuss this issue.

Authors’ Response: We thank the reviewer for highlighting this important topic. We have added a new section on this in the review. See section 4 in the revised paper “Oncologic Safety of Fertility-Preservation Surgeries in Different Histologic Types”

Reviewer 2 Report

Comments and Suggestions for Authors

A systematic review and summary of indications and treatment strategies for fertility preservation in cervical cancer are presented in this manuscript. The author possesses a sufficient understanding of the present state of cervical cancer treatments that preserve fertility. The authors adopt an analytical stance in their investigation of various treatment modalities in this work. Standard format and language style coordination for the paper's writing. Although the number of references within five years has exceeded 50%, the quality of the references still needs further improvement.

Author Response

A systematic review and summary of indications and treatment strategies for fertility preservation in cervical cancer are presented in this manuscript. The author possesses a sufficient understanding of the present state of cervical cancer treatments that preserve fertility. The authors adopt an analytical stance in their investigation of various treatment modalities in this work. Standard format and language style coordination for the paper's writing. Although the number of references within five years has exceeded 50%, the quality of the references still needs further improvement.

Authors’ Response: We thank the reviewer for this comment. We have edited some of the references and added more recent publications. See examples in the revised paper ref # 4, 25, 35

Reviewer 3 Report

Comments and Suggestions for Authors

Dear authors,

thank you for the opportunity to review this paper.  It is a narrative review of the literature on fertility sparing treatment strategies in cervical cancer and its possible indications. 

I have some consideration:

-        First of all, I ask what were the search filters for the selection of articles to be included in the review? Indeed, I note at a glance that the literature lacks some systematic reviews that provide important data on the subject:

o   Nehzat et al. 2020, Reproductive and oncologic outcomes after fertility-sparing surgery for early stage cervical cancer: a systematic review

-        Lines 42-44: the sentence is not clear and the reference is missing.

-        In the “introduction” section improve the risk factor and possibility of the role of sentinel linphonode dissection.

-        Line 76: it is important to explain in detail the results of LACC trial e the role of MIS in case of small tumor.

-        Is there a role of robotic surgery?

  • Lines 136-137: There is recent paper that describe the role of LVSI, could you add it? (PMID: PMID: 34116834)

In the literature there are similar works such as that of Zaccarini at el. “Cervical Cancer and Fertility-Sparing Treatment”. However, your work, unlike those present, also deals with issues such as ovarian transposition and the cryopreservation of oocytes, embryos and ovarian tissue.

The review could be supplemented by comparing the success rate in terms of pregnancy between the various techniques described.

Comments on the Quality of English Language

Moderate editing of English language required

Author Response

Dear authors,

thank you for the opportunity to review this paper.  It is a narrative review of the literature on fertility sparing treatment strategies in cervical cancer and its possible indications. 

I have some consideration:

       First of all, I ask what were the search filters for the selection of articles to be included in the review? Indeed, I note at a glance that the literature lacks some systematic reviews that provide important data on the subject:

Nehzat et al. 2020, Reproductive and oncologic outcomes after fertility-sparing surgery for early stage cervical cancer: a systematic review

Authors’ response: We thank the reviewer for this comment. As was mentioned this was a narrative review and not a systematic review. The articles cited in the papers were chosen according to relevance and importance of the study. As we did not do a systematic review, some important papers, such as the one recommended by the reviewer were missed. We added the recommended paper and its results to the text (See lines 158-166, reference 30).

        Lines 42-44: the sentence is not clear and the reference is missing.

Authors’ response: We have edited the sentence and added references.

       In the “introduction” section improve the risk factor and possibility of the role of sentinel linphonode dissection.

Authors’ response: We added a section on lymph node assessment and role of sentinel LND under the “radical trachelectomy” section, following the indications for radical trachelectomy. See lines 57-69

       Line 76: it is important to explain in detail the results of LACC trial e the role of MIS in case of small tumor.

Authors’ response: We thank the reviewer for this comment. We have expanded on the LACC trial and the limitation of this study in regards to small tumors. See lines 93-99

 Is there a role of robotic surgery?

Authors’ response:  Some of the studies that we included (e.g., Smith et al 2020, Nezhat et al 2020) included robotic surgeries in the MIS group. There is not enough robust data on robotic surgeries for fertility-preservation treatments in cervical cancer. Most studies combine it with laparoscopy as one group. Therefore, we did not expand on it in our review.

Lines 136-137: There is recent paper that describe the role of LVSI, could you add it? (PMID: PMID: 34116834)

Authors’ response: LVSI is an important risk factor for recurrence and determining adjuvant treatment. We did not comment on the role of LVSI in isolation in this paper as it alone does not impact eligibility for fertility preservation treatments.

In the literature there are similar works such as that of Zaccarini at el. “Cervical Cancer and Fertility-Sparing Treatment”. However, your work, unlike those present, also deals with issues such as ovarian transposition and the cryopreservation of oocytes, embryos and ovarian tissue.

The review could be supplemented by comparing the success rate in terms of pregnancy between the various techniques described.

Authors’ response: We added a table comparing the different ARTs with respect to indications and pregnancy outcomes. See table 2 in the revised paper.

Reviewer 4 Report

Comments and Suggestions for Authors

This review article focus on Fertility Preservation in Cervical Cancer. This article is very important topic and well written about the method of fertility preservation. However, this manuscript should be improved to provide more new information.  

Fertility preservation in cervical cancer is important topic but you should discuss new topic, such as uterine transplantation and fertility preservation in advanced stage cervical cancer.

What is your methodology to review this manuscript?  You should clarify if you used PRISMA Statement.

In Table 1, you summaries several studies evaluating neoadjuvant chemotherapy followed by fertility sparing surgery for cervical cancer. In this table, you should add the information about stage before and after chemotherapy.

You included the information about ovarian transposition. I recommend that you make the table about successful delivery after ovarian transposition.

Moreover, you stated that cryopreservation of oocytes, embryos and ovarian tissue is performed for cervical cancer. Actually, in this review you should show some successful live birth article. You only show unsuccessful live birth article of ovarian tissue cryopreservation due to uterine fibrosis. 

Author Response

This review article focus on Fertility Preservation in Cervical Cancer. This article is very important topic and well written about the method of fertility preservation. However, this manuscript should be improved to provide more new information.  

Fertility preservation in cervical cancer is important topic but you should discuss new topic, such as uterine transplantation and fertility preservation in advanced stage cervical cancer.

Authors’ response: We thank the reviewer for this comment. We addressed fertility-preservation in advanced stage in sections 6 and 7 in the revised manuscript. In these sections we discuss ovarian transposition and ART in patients with advanced disease that are not eligible for fertility-sparing surgery. As for the uterine transplantation, while it is extremely interesting approach for younger patients with advanced rectal cancers, however, there is not a lot of data to discuss the oncologic safety and reproductive outcome and therefore we did not include that in our paper. 

What is your methodology to review this manuscript?  You should clarify if you used PRISMA Statement.

Authors’ response:  We provided a review article on fertility preservation options in cervical cancer. This was not a systematic review and therefore we did not use the PRISMA statement. The articles cited in the papers were chosen according to relevance and importance of the study.

In Table 1, you summaries several studies evaluating neoadjuvant chemotherapy followed by fertility sparing surgery for cervical cancer. In this table, you should add the information about stage before and after chemotherapy.

Authors’ response: We thank the reviewer for highlighting this. We edited table 1 and added stage of disease and tumor size prior to chemotherapy. The information on tumor size after chemotherapy was not consistent between studies and therefore, we did not include that in the table.

You included the information about ovarian transposition. I recommend that you make the table about successful delivery after ovarian transposition.

Authors’ response: Data on successful deliveries after ovarian transposition is scarce and is greatly affected by selection and publication bias. In addition, this manuscript is focused primarily on fertility preservation. While successful delivery is the end goal, it was not the primary focus.

Moreover, you stated that cryopreservation of oocytes, embryos and ovarian tissue is performed for cervical cancer. Actually, in this review you should show some successful live birth article. You only show unsuccessful live birth article of ovarian tissue cryopreservation due to uterine fibrosis. 

Authors’ response: This manuscript is focused on fertility preservation and not fertility outcome. Without going into depth, we added some data on livebirth rates following oocyte and embryo cryopreservation. See lines 283-290 

Round 2

Reviewer 1 Report

Comments and Suggestions for Authors

Questions/comments are properly addressed

Reviewer 3 Report

Comments and Suggestions for Authors

thank you for your response to the different questions

Comments on the Quality of English Language

Minor editing of English language required

Reviewer 4 Report

Comments and Suggestions for Authors

Thank you  for giving me an oportunity to revew this manuscript. Author addressed to my comments.